# Contemporary Cardiovascular Risk Assessment for Type 2 Diabetes Including Heart Failure as an Outcome: The Fremantle Diabetes Study Phase II

**DOI:** 10.3390/jcm9051428

**Published:** 2020-05-11

**Authors:** Wendy A. Davis, Valentina Hellbusch, Michael L. Hunter, David G. Bruce, Timothy M. E. Davis

**Affiliations:** 1Medical School, The University of Western Australia, Fremantle Hospital, PO Box 480, Fremantle, WA 6959, Australia; wendy.davis@uwa.edu.au (W.A.D.); peugeot404au@yahoo.com.au (V.H.); david.bruce@uwa.edu.au (D.G.B.); 2Busselton Population Medical Research Institute, Busselton, WA 6280, Australia; michael.hunter@uwa.edu.au

**Keywords:** type 2 diabetes, cardiovascular disease, risk prediction, competing risk regression

## Abstract

*Background:* Type 2 diabetes (T2D) cardiovascular disease (CVD) risk assessment has limitations. The aim of this study was to develop a risk equation adding heart failure (HF) to conventional major adverse cardiovascular events (MACE, myocardial infarction, stroke, and CVD death) and allowing for non-CVD death. *Methods:* 1551 community-based people with T2D (mean age 66 years, 52% males) were followed from baseline in 2008–2011 for five years to the first CVD event/death. Cox and competing risk regression identified predictors of three-point MACE and four-point MACE (including HF). Discrimination was assessed by the area under the receiver-operating characteristic curve (AUC) and calibration by the Hosmer-Lemeshow test. Sensitivity, specificity, positive predictive value (PPV), and negative predictive value (NPV) were determined for a 10% five-year CVD risk cut-off. *Results:* 143 participants (9.2%) experienced a three-point MACE during 7,111 person-years of follow-up and 245 (15.8%) a four-point MACE during 6,896 person-years. The best model was the competing risk four-point MACE (221 predicted events (14.3%), AUC 0.82 (95% CI: 0.79–0.85), Hosmer-Lemeshow test, *p* = 0.17, sensitivity 79.2%, specificity 68.1%, PPV 31.8%, NPV 94.6%) with validation in 177 adults with T2D from an independent population (AUC 0.81 (0.74–0.89). *Conclusions:* A validated four-point MACE competing risk model reliably predicts key T2D CVD outcomes.

## 1. Introduction

Risk equations have been used to predict cardiovascular disease (CVD) events that facilitate clinical management for more than four decades [1]. There are currently >100 such equations and 45 specific equations for type 2 diabetes [2]. Reflecting differences in populations, endpoint ascertainment, the predictive variables included and algorithms employed, and changes in management over time, there is considerable variability in their characteristics and performance, while the equations may not perform well when used outside the populations in which they were developed [3]. There is, therefore, an argument that locally developed equations should ideally be used, but there may not be relevant contemporary longitudinal data available to allow this.

In Australia, for example, the CVD risk assessment recommended by the National Vascular Disease Prevention Alliance is based on the Framingham equation from the US, which includes diabetes yes/no as its only diabetes-specific variable [4]. We assessed this equation using data from participants with type 2 diabetes in the representative community-based Fremantle Diabetes Study Phase I (FDS1) cohort followed from recruitment between 1993 and 1996 and found poor discrimination (the ability to predict who will have a future event) and calibration (the agreement between overall predicted and observed event rates) [5]. The United Kingdom Prospective Diabetes Study (UKPDS) calculator, which is specific for type 2 diabetes and based on an intervention trial conducted between 1977 and 1997 [6,7], estimated double the coronary heart disease (CHD) event rate than what occurred in the FDS1 cohort, with modest discrimination and poor calibration, even if stroke risk prediction was acceptable [5].

In response to the potentially misleading information generated by the Framingham and UKPDS equations applied to Australians with type 2 diabetes [5], we utilised FDS1 data to develop an equation predicting the five-year risk of hospitalisation with the major adverse cardiovascular events (MACE) myocardial infarction (MI), stroke, and CVD death (from cardiac/cerebrovascular causes, or sudden death; three-point MACE) [8]. This equation was validated using contemporaneous Busselton Health Study (BHS) data from people with type 2 diabetes in a semi-rural population with good discrimination, goodness-of-fit, and accuracy, and positive predictive value (PPV) and negative predictive value (NPV) [8], and also in adults with type 1 diabetes from FDS1 [9]. When assessed against other diabetes-specific CVD risk calculators, its *c*-index or area under the receiver operating characteristic curve (AUC) was among the highest in a meta-analysis [10], and it performed comparably in population-based cohorts with type 2 diabetes from the Netherlands [3], Germany [3], and Scotland [11].

A criticism of the UKPDS calculator has been that changes in type 2 diabetes diagnosis, management, and outcome since its cohort was enrolled in the 1970s and 1980s mean that it should not be used in contemporary populations [3,5]. Similarly, there have been changes in CVD risk management since FDS1 was conducted, including increased use of cardiovascular pharmacotherapies informed by relevant intervention trials [12]. The competing risk of death from non-CVD causes has increased as a result [13]. A lowering of the fasting serum glucose diagnostic criterion in 1999 and greater awareness of the importance of early diabetes detection and management to prevent complications have resulted in fewer people with clinically significant glucose intolerance remaining undiagnosed [14]. While the three-point MACE remains the conventional endpoint in intervention trials and risk calculation in type 2 diabetes, heart failure (HF) has emerged as an additional key outcome [15]. Stand-alone risk prediction equations for HF hospitalisation complicating type 2 diabetes have been developed [16].

Given these considerations, the availability of detailed longitudinal data from a representative sample of people with type 2 diabetes in FDS Phase II (FDS2) recruited between 2008 and 2011, 15 years after but from the same catchment as FDS1 [17], and from the Busselton Diabetes Study (BDS) conducted in an independent community using the same study methodology as, and contemporaneously with, FDS2 [18], we have i) assessed the performance of the FDS1 five-year CVD risk equation in the FDS2 cohort, ii) assessed the performance of the only other diabetes-specific CVD risk equation with significant Australian representation, derived from the Action in Diabetes and Vascular Disease: Preterax and Diamicron Modified Release Controlled Evaluation (ADVANCE) trial [19] in which 1 in 7 participants were Australasian [20], and iii) developed and validated a new CVD risk equation utilising longitudinal FDS2 data, which included HF as an endpoint and allowed for the competing risk of death from non-CVD causes.

## 2. Materials and Methods

### 2.1. Participants, Epidemiological Setting, and Approvals

In both FDS phases, people with known diabetes were recruited from a postcode-defined geographical area with 157,000 residents that surrounds the port city of Fremantle in the state of Western Australia (WA). Socio-economic data from this area at the time of FDS2 recruitment showed an average Index of Relative Socio-economic Advantage and Disadvantage [21] of 1033 with a range by postcode of 977–1113, and figures paralleling the Australian national mean ± SD (1000 ± 100). Of 4639 residents with diabetes identified between 2008 and 2011, 1668 (36%) were recruited to FDS2. Sixty-four former FDS1 participants who had moved out of the area were also recruited, giving a total of 1732 of whom 1551 had clinically-defined type 2 diabetes. Descriptions of recruitment, sample characteristics, and details of non-recruited people with diabetes have been published [17]. The FDS2 was approved by the Human Research Ethics Committee of the Southern Metropolitan Area Health Service (07/397 18 October 2007).

### 2.2. Clinical and Laboratory Methods

Participants had comprehensive FDS2 face-to-face assessments at baseline and biennially for up to six years [17]. At each visit, demographic and clinical information was documented, and a physical examination and associated investigations were performed. Fasting blood and first-morning urine samples were obtained for standard biochemical tests done in a single nationally accredited laboratory [22], with between-run imprecision for all assays <3.5% except for urine albumin and serum HDL-cholesterol (<5.0%). A Body Shape Index (ABSI) incorporating waist circumference (WC), body mass index (BMI), and height was calculated as WC/(BMI^2/3^ × height^1/2^) [23].

Chronic complications were identified using standard criteria [22], including peripheral sensory neuropathy (PSN, a score >2/8 on the clinical portion of the Michigan Neuropathy Screening Instrument), retinopathy (any grade detected by retinal photography and/or ophthalmologist assessment), nephropathy (based on urinary albumin:creatinine ratio (uACR)), renal impairment (based on estimated glomerular filtration rate (eGFR) using the Chronic Kidney Disease Epidemiology Collaboration (CKD-EPI) equation), CHD (self-reported history of, or prior hospitalisations with, MI, angina, and/or revascularisation), cerebrovascular disease (self-reported history of, or prior hospitalisations with, stroke/transient ischaemic attack), and peripheral arterial disease (PAD; ankle:brachial index ≤ 0.90 on either leg or diabetes-related amputation).

### 2.3. Ascertainment of Incident Myocardial Infarction, Stroke, and Heart Failure

Collection of FDS2 participant morbidity and mortality data continues through health service linkages using the WA Data Linkage System (WADLS) [24], as approved by the WA Department of Health Human Research Ethics Committee. The Hospital Morbidity Data Collection (HMDC) includes comprehensive data on all public/private hospitalisations in WA. Using International Classification of Disease (ICD)-10-AM coding, the HMDC was searched for MI, stroke, and HF from study entry to end-2016. Mortality was identified from the Registry for Births, Deaths, and Marriages. Causes of death, based on the death certificate or coroner’s determination, were classified independently under the UKPDS system [25] by two study physicians (DGB, TMED). In the case of discrepancies, case notes were consulted and a consensus was obtained. FDS2 and linked outcome data were used for two composite endpoints, which include the conventional three-point MACE and the three-point MACE plus hospitalisation for/with HF (four-point MACE).

### 2.4. Validation Dataset

The BDS is an observational study of people with known diabetes and age-matched and sex-matched normoglycemic residents from Busselton, a shire with 31,000 people in south-west WA, recruited between 2009 and 2010 [18]. Participants were identified from prior involvement in community-based BHS studies that commenced in 1966. Residents with diabetes in BHS surveys conducted in 1994–1995 and 2005–2007 were invited to participate in the BDS, with additional recruitment through health professional referral, word of mouth, and advertising. World Health Organisation/International Diabetes Federation recommendations [26] were used to verify diabetes/normoglycemia ascertained from self-report/fasting serum glucose concentrations in the BHS surveys. The time to first MI, stroke, or HF hospitalisation over the next five years was determined using the HMDC and Death Registrations, as described for the FDS2 cohort.

### 2.5. Statistical Analysis

The computer packages IBM SPSS Statistics 25 (IBM Corporation, Armonk, NY, USA) and StataSE 15 (StataCorp LP, College Station, TX, USA) were used for statistical analysis. Data are presented as percentages, mean ± SD, geometric mean (SD range), or when variables did not conform to a normal or log-normal distribution, median, and inter-quartile range. For independent samples, two-way comparisons for proportions were by Fisher’s exact test, for normally distributed variables by the Student’s *t*-test, and for non-normally distributed variables by the Mann-Whitney *U*-test. One in 24 (4.1%) participants had missing data for one or more variables, which were multiply imputed (×20). With the exception of baseline characteristics and the assessment of the performance of the FDS1 five-year and ADVANCE four-year CVD risk equations for type 2 diabetes in the FDS2 type 2 diabetes cohort, results are reported for imputed data.

Both Cox and Fine and Gray competing risk regression modelling [27] were used to determine independent predictors of time to the first three-point MACE and four-point MACE. Variables were included if they were clinically plausible and likely to be routinely available with a bivariable *p* < 0.20. They were removed one at a time, those with least statistical significance first, until all variables in the model were significant at *p* < 0.05. Excluded variables were added again to the final model to confirm lack of significance, clinical importance, or confounding. The proportional hazards assumption was checked using the time-dependent covariate method for each independent risk factor separately.

Since the baseline cumulative hazard refers to a state in which all variables are zero, and zero age, heart rate, and HbA_1c_, in particular, are nonsensical in this context, continuous variables were centered at the respective mean cohort values. Regression coefficients from the final models were used to compute a linear risk function, L, that uses a participant with average values of each risk factor as reference. The subsequent result was exponentiated to calculate a five-year CVD event probability, P, after insertion into a survival function P = 1 − exp(−c*exp(L)), where c is the five-year baseline cumulative hazard for the Cox model or the baseline cumulative sub-distribution hazard for the competing risk model [28].

Discrimination was assessed by the area under the receiver-operating characteristic curve (AUC). Calibration was assessed using the Hosmer-Lemeshow test. Accuracy was assessed by the Brier score (mean squared error, range 0–1, the lower the better), and sensitivity, specificity, PPV, and NPV were determined for a 10% five-year CVD risk cut-off (or 8% for the ADVANCE four-year CVD risk equation).

## 3. Results

### 3.1. Baseline Characteristics of FDS1 Versus FDS2 Type 2 Diabetes Cohorts

The baseline characteristics of the FDS1 and FDS2 type 2 diabetes cohorts are summarised in Table 1. Compared with FDS1 participants, those in FDS2 were not significantly different in terms of sex distribution, but were older at study entry and younger at diagnosis of diabetes with a consequently longer median diabetes duration. The FDS2 cohort was significantly more obese, drank more alcohol but smoked less, was better educated, and more likely to speak English fluently. Those in FDS2 had lower mean systolic blood pressure, glycaemia, serum total cholesterol, triglycerides, and uACR. The prevalence of CHD and cerebrovascular disease were not significantly different between phases, but the prevalence of PAD was less in FDS2 than FDS1 while that of PSN was greater. At the end of five years of follow-up, 14.6% of the FDS1 cohort had died versus 12.0% of the FDS2 cohort (*p* = 0.044).

### 3.2. Performance of the FDS1 Five-Year CVD Risk Equation in the FDS2 Type 2 Diabetes Cohort

During follow-up from the FDS2 baseline assessment to first incident three-point MACE or death from other causes or census (at 5.0 years), whichever came first, a total follow-up of 7018 person-years (4.6 ± 1.1 years), 138 (9.1% (95% CI: 7.7–10.7%)) of 1522 (98.1%) participants in the FDS2 cohort with complete data had a three-point MACE compared with 251 (16.5%) predicted by the FDS1 five-year CVD risk equation. Discrimination was moderate (AUC (95% CI) 0.72 (0.67–0.76), *p* < 0.001) and accuracy (Brier score 0.09 (0–0.79)) good, but calibration poor (Hosmer-Lemeshow test, *p* < 0.001). At a risk cut-off of 10%, the sensitivity was 77.5%, specificity was 51.4%, PPV was 13.7%, and NPV was 95.8%.

### 3.3. Performance of the ADVANCE CVD Risk Equation in the FDS2 Type 2 Diabetes Cohort

The ADVANCE four-year CVD risk equation can only be applied to those without known CVD at baseline, and 513 (33.1%) FDS2 participants with type 2 diabetes were consequently excluded from this validation. During follow-up to the first three-point MACE, death from other causes or census (at 4.0 years), whichever came first, a total of 3978 person-years (3.8 ± 0.6 years), 38 (3.8% (95% CI: 2.7–5.2%)) of 1001 (96.4%) eligible FDS2 participants had a three-point MACE when compared with 36 (3.6%) predicted by the ADVANCE four-year CVD risk equation. Discrimination was moderate (AUC (95% CI) 0.67 (0.58–0.76); *p* < 0.001), accuracy good (Brier score 0.04 (0–0.997)), and calibration (Hosmer-Lemeshow test, *p* = 0.08) acceptable. At a risk cut-off of 8%, the sensitivity was 26.3%, specificity was 90.9%, PPV was 10.2%, and NPV was 96.9%.

### 3.4. Participant Characteristics and Outcome

The bivariable baseline associates of the five-year incident three-point MACE and four-point MACE are shown in Appendix A. For 7335 person-years (4.7 ± 0.9 years) of follow-up to death or census (at 5.0 years), whichever came first, 186 (12.0%) died from any cause of whom 49 (26.3%) died from cardiac or cerebrovascular causes, or suffered a sudden death.

During 7,111 person-years of follow-up to the first three-point MACE or death or census, whichever came first, 143 (9.2% (95% CI: 7.9–10.8%)) participants experienced a three-point MACE. These participants were older at study entry, more likely to be Aboriginal Australians and current smokers, but less likely to be married or physically active in the preceding week than those who did not have an event. They had longer diabetes duration with more intensive blood glucose-lowering treatment but higher HbA_1c_ and were more likely to have had severe hypoglycaemia. They had a higher ABSI, systolic blood pressure, pulse pressure, and heart rate, and were more likely to be on antihypertensive medications and aspirin, and to have orthostatic hypotension, atrial fibrillation (AF), and left ventricular hypertrophy (LVH) based on electrocardiographic criteria. A history of CVD, microvascular complications, and chronic kidney disease were more prevalent in those with incident three-point MACE. Serum lipid levels and use of lipid-modifying medications were not associated with an incident three-point MACE.

During 6896 person-years of follow-up to the first incident four-point MACE endpoint or death or census, whichever came first, 245 (15.8% (14.0–17.7%)) participants experienced a four-point MACE. The statistically significant baseline associates of incident four-point MACE were similar to those for three-point MACE with the addition of sex and education beyond the primary level, and the exclusion of orthostatic hypotension.

### 3.5. Independent Associates of First Incident Three-point MACE

Model 1: In Cox regression modelling, independent risk factors for the first three-point MACE were age (centred at 65.7 years), (age − 65.7)^2^, being Aboriginal Australian, ln(HbA_1c_) (centred at 3.98), ln(serum total:HDL-cholesterol ratio) (centred at 1.27), ln(urinary albumin:creatinine ratio) (centred at 1.22), eGFR 45–59, and <60 mL/min/1.73m^2^, LVH, history of CVD, and presence of PAD. Variables considered for entry but excluded due to statistical non-significance included marital status, physical activity, smoking, diabetes treatment, history of severe hypoglycaemia, ABSI, systolic blood pressure, pulse pressure, orthostatic hypotension, heart rate, antihypertensive use, aspirin use, AF, PSN, retinopathy, and history of HF. Inspection of the time-dependent covariate for each independent risk factor showed no violation of the proportional hazards assumption (*p* > 0.11). The regression coefficients for each risk factor are presented in Table 2 (baseline cumulative hazard, c = 0.0201).

Model 2: In Fine and Gray competing risk regression modelling, independent risk factors for the first three-point MACE were the same as for Model 1 excluding ln(serum total:HDL-cholesterol ratio) (see Table 2; baseline cumulative sub-distribution hazard, c = 0.0200). There was no violation of the proportional hazards assumption (*p* > 0.13).

### 3.6. Independent Associates of First Incident Four-point MACE

Model 3: In Cox regression modelling, the independent risk factors for first four-point MACE were the same as for Model 1 with the addition of diabetes duration and history of HF, and the exclusion of eGFR 45–59 mL/min/1.73m^2^ and ln(serum total: HDL-cholesterol ratio) (see Table 2; baseline cumulative hazard, c = 0.0573). There was no violation of the proportional hazards assumption (*p* > 0.10).

Model 4: In Fine and Gray modelling, independent risk factors for first four-point MACE were the same as for Model 3 with the addition of sex and heart rate (see Table 2; baseline cumulative sub-distribution hazard, c = 0.0478). There was no violation of the proportional hazards assumption (*p* > 0.12). As an example, a 58.7-year-old non-Aboriginal Australian man with diabetes duration 10.0 years, HbA_1c_ 51 mmol/mol, heart rate 58 beats/minute, uACR 0.54 mg/mmol, eGFR ≥ 45 mL/min/1.73m^2^, PAD, but no LVH, no HF, and no CVD, would have a probability of an incident four-point MACE during the next five years of 4.5% (see Appendix A). By comparison, for a 45.6-year-old Aboriginal Australian woman with diabetes duration 13.0 years, HbA_1c_ 100 mmol/mol, heart rate 63 beats/minute, uACR 25.0 mg/mmol, eGFR ≥ 45 mL/min/1.73m^2^, no PAD, but LVH, HF, and CVD. The probability would be 81.7%.

### 3.7. Model Performance

The performances of Models 1-4 are shown in Table 3. Figure 1 illustrates the calibration for each model. Only the four-point MACE models had acceptable calibration with Model 4 superior to Model 3 (Hosmer-Lemeshow test, *p* = 0.17 and 0.058, respectively). Model 4 predicted a mean five-year four-point MACE probability of 14.3% compared with the observed 15.8% (95% CI: 14.0–17.7%). Discrimination (AUC = 0.82 (95% CI: 0.79–0.85), *p* < 0.001) and accuracy (Brier score = 0.10 (range: 0–0.95)) were good. For a five-year predicted four-point MACE cut-off of 10%, sensitivity was 79.2%, specificity was 68.1%, PPV was 31.8%, and NPV was 94.6%.

### 3.8. External Validation

The baseline characteristics of the FDS2 and BDS validation cohorts are compared in Appendix A. The sex distribution, diabetes duration, CVD history, heart rate, renal function, and proportions with chronic kidney disease were not statistically different, but the validation cohort was significantly older, none was Aboriginal, and the percentage with LVH was higher and PAD lower. During five years of follow-up, 37 (20.9% (95% CI: 15.3–27.8%)) of the 177 validation cohort with complete data had a four-point MACE compared with a mean risk of 12.5% (22 cases) predicted by Model 4. There was good discrimination (AUC = 0.81 (0.74–0.89), *p* < 0.001), calibration (Hosmer-Lemeshow test, *p* = 0.16), and accuracy (Brier score = 0.15 (0–0.92)). At a risk cut-off of 10%, sensitivity was 75.7%, specificity was 73.6%, PPV was 43.1%, and NPV was 92.0%.

## 4. Discussion

We utilised detailed longitudinal data from a community-based cohort of people with type 2 diabetes to develop and validate a CVD risk equation that could be used to guide the nature and intensity of contemporary risk factor management. When the FDS1 equation was developed from data collected largely in the 1990s, it had good discrimination for three-point MACE (AUC 0.80) [8]. The need for an updated version is illustrated by the lower AUC (0.72) and poor calibration when it was applied to outcome data from the more recently recruited FDS2 cohort, which differed from the FDS1 cohort in a number of key variables relevant to CVD-associated endpoints. The FDS2 three-point MACE equation AUC was higher (0.77) and, when HF was added as a fourth composite endpoint, the AUC was 0.81 and 0.82 for the Cox and competing risk multivariable models, respectively. Calibration and accuracy were also good for the four-point MACE competing risk model, which had the best results overall of all four models that were tested for sensitivity, specificity, PPV, and NPV. It also performed well in an independent cohort of people with type 2 diabetes from another Australian population center (AUC 0.81). The ADVANCE equation [19] was restricted to people without known CVD at baseline and had moderate discrimination in the FDS2 cohort (AUC 0.67). This may, as in the case of the UKPDS calculator [29], reflect its development in the context of a clinical trial involving participant selection and protocol-driven management, which would question its generalisability.

The present findings suggest that the FDS2 four-point MACE competing risk equation is the preferred option for routine use when compared with the FDS1 and ADVANCE equations, and the other three derived from FSD2 data. Although developed in an Australian population, the performance of the original three-point MACE FDS1 risk equation in relation to the others available, at least for independent Europid populations [3,11], suggests that its preferred four-point MACE successor, derived 15 years later but from people from the same geographic area, could also be applied to the equation outside Australia. Nevertheless, the high CVD risk of Aboriginal Australians [30] has been captured as one of its variables, as shown by the very high five-year risk of a CVD/HF event (81.7%) in an Aboriginal woman relative to that in an older Caucasian male (4.5%) in the examples provided. The other variables in this model are all plausible given their independent associations with adverse CVD outcomes in other studies [31,32].

We considered only variables that were readily available in primary care. This was, in part, to ensure broad-based utility, but also because novel biomarkers and specialised measurements such as serum high-sensitivity troponin T, plasma N-terminal pro-B-type natriuretic peptide, plasma high sensitivity C-reactive protein, and coronary artery calcification score await confirmation of a clinically relevant contribution to risk prediction over and above, or instead of, conventional risk factors in type 2 diabetes [33]. In the present study, use of conventional clinical/laboratory variables produced good discrimination and other metrics such as NPV, which was >94% in all four models. This suggests that additional variables would not add substantively to model performance.

There were some plausible risk factors that were not included in the FDS2-based regression models we developed. Smoking was also not a significant predictor of CVD risk in the FDS1 equation, a finding attributed to the low percentage of current smokers (15.1% [9] versus 34% and 25% of UKPDS male and female subjects, respectively [7]). This figure had fallen to 10.5% in FDS2, while the inclusion of PAD in the models may, in part, have reflected its close association with smoking. Systolic blood pressure was also in neither the FDS1 risk equation [9] nor in the four FDS2 models. This might be explained by the availability of LVH, but it could also reflect effective use of antihypertensive therapy and the non-linear J-shaped relationship between blood pressure and outcome [34]. In addition, the inclusion of prior CHD and/or cerebrovascular disease is likely to have masked a contribution of risk factors such as hypertension and dyslipidaemia. There is an argument that prior CVD implies a very high future CVD risk on its own, and, thus, the need for the most intensive management regardless of risk estimates. However, unlike the ADVANCE risk equation [19], we considered that knowledge of the likelihood of future events in people with type 2 diabetes already diagnosed with CVD may still aid patient education, and compliance with lifestyle changes and pharmacotherapy.

There has been a trend for developing simultaneous risk equations for individual CVD outcomes in type 2 diabetes [35], or for stand-alone CVD complications such as MI [7], stroke [6,36], and HF [16]. We recommend that a global CVD risk score that predicts the combination of the main causes of morbidity and mortality in type 2 diabetes should be prioritised. It underlies a simple clinical strategy that should be easier to understand for people with diabetes than a series of scores that generate complication-specific management. MACE components including HF [37] are manifestations of, or have a major contribution from, atherosclerosis, and so there is likely to be significant overlap between the risk factors for individual complications including hyperglycaemia in the case of HF due to diabetic cardiomyopathy [38] rather than atherosclerosis.

A competing risk approach has started to emerge when prediction models are being developed [39,40]. Its main advantage is that ignoring competing risks can lead to estimates of predicted risk using Cox regression that are biased upwards [41]. Such overestimates may increase the likelihood of inappropriate management with implications for cost, and treatment-related interactions and adverse effects. The incidence of CVD events in many Western populations is declining [42], and so competing risks may become of greater concern to risk prediction in the future.

The present study had limitations. Although the FDS2 cohort was from a typical urban Australian population base and were representative of those with the disease in the study catchment area, their characteristics may differ from those of people with type 2 diabetes in other geographical settings. We used hospitalisations and deaths as key outcomes, which would not have captured silent events or symptomatic events that were managed outside hospital. The strengths of the study include the detailed participant-level data and access to a validated data linkage system as well as the availability of a contemporaneous validation cohort (the BDS).

## 5. Conclusions

Data and CVD risk equations from FDS1 and FDS2, including previous validations of the Framingham and UKPDS equations [5] and the present ADVANCE risk calculator validation, show that there is a need, where the data are available, to develop local CVD risk prediction equations. These equations should be updated as diabetes epidemiology changes. This includes consideration of clinically significant events such as HF, which may be increasing in incidence. Where local data are unavailable or unsuitable, equations developed in other ethnically similar populations can be applied as long as their potential limitations are acknowledged and the possibility of recalibration is explored [11]. The four-point MACE competing risk equation developed from FDS2 data and validated in the independent BDS cohort could be considered as a replacement for the Framingham equation-based CVD risk assessment currently recommended for Australians with type 2 diabetes [4]. The possibility of its application beyond an Australian context should be explored.

## Figures and Tables

**Figure 1 jcm-09-01428-f001:**
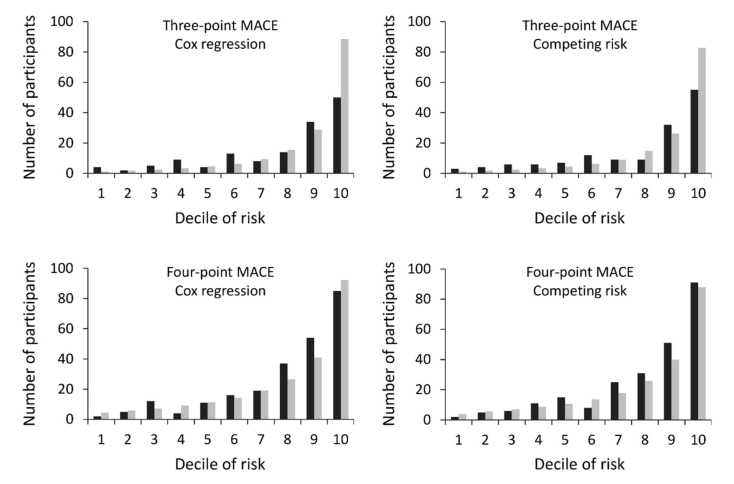
Observed versus predicted numbers of incident first three-point major adverse cardiovascular events (MACE) and four-point MACE by decile of risk in the Fremantle Diabetes Study Phase II type 2 diabetes cohort in Models 1–4. Black bars represent observed events and grey bars represent predicted events.

**Table 1 jcm-09-01428-t001:** Baseline characteristics of type 2 diabetes participants by Fremantle Diabetes Study Phase. Data are percentages, mean ± SD, geometric mean (SD range), or median (inter-quartile range).

	Phase I	Phase II	*p*-Value
Number (%)	1296	1551	
Age (years)	64.0 ± 11.3	65.7 ± 11.6	<0.001
Sex (% male)	48.6	51.9	0.08
ApoE ε4 genotype (%)	21.8	23.6	0.27
Ethnic background (%)			<0.001
Anglo-Celt	63.3	53.4	
Southern European	18.4	12.6	
Other European	8.5	7.2	
Asian	3.3	4.4	
Aboriginal	1.3	6.7	
Mixed/other	5.2	15.7	
Not fluent in English (%)	15.3	10.6	<0.001
Currently married/*de facto* relationship (%)	65.7	62.7	0.10
Educational attainment beyond primary level (%)	74.0	86.7	<0.001
Smoking status (%)			0.001
Never	44.7	45.5	
Ex	40.2	44.0	
Current	15.1	10.5	
Alcohol consumption (standard drinks/day)	0 [0–0.8]	0.1 [0–1.2]	<0.001
Age at diabetes diagnosis (years)	57.9 ±11.7	55.5 ± 12.3	<0.001
Diabetes duration (years)	4.0 [1.0–9.0]	9.0 [3.0–15.8]	<0.001
Diabetes treatment (%)			<0.001
Diet	31.9	24.1	
Oral hypoglycaemic agents (OHAs)/non-insulin injectables	55.7	53.4	
Insulin only	9.5	5.9	
Insulin + OHAs/non-insulin injectables	2.8	16.6	
Fasting serum glucose (mmol/L)	8.3 (5.9–11.5)	7.6 (5.6–10.2)	<0.001
HbA_1c_ (%)	7.3 (5.9–9.2)	7.1 (5.9–8.5)	<0.001
HbA_1c_ (mmol/mol)	56 (41–77)	54 (41–69)	<0.001
BMI (kg/m^2^)	29.6 ± 5.4	31.2 ± 6.1	<0.001
Central obesity (by waist circumference, %)	64.5	71.4	<0.001
ABSI (m^11/6^kg^−2/3^)	0.082 ± 0.005	0.081 ± 0.005	0.18
Systolic blood pressure (mmHg)	151 ± 24	146 ± 22	<0.001
Diastolic blood pressure (mmHg)	80 ± 11	80±12	0.51
Taking antihypertensive medication (%)	50.9	73.7	<0.001
Total serum cholesterol (mmol/L)	5.4 (4.4–6.5)	4.2 (3.3–5.4)	<0.001
Serum HDL-cholesterol (mmol/L)	1.01 (0.75–1.38)	1.19 (0.92–1.55)	<0.001
Total:HDL-cholesterol ratio	5.3 (3.8–7.4)	3.5 (2.6–4.8)	<0.001
Serum triglycerides (mmol/L)	2.2 (1.2–3.9)	1.5 (0.9–2.6)	<0.001
Taking lipid-lowering medication (%)	10.5	68.5	<0.001
Taking aspirin (%)	22.0	37.5	<0.001
Cerebrovascular disease (%)	10.0	11.4	0.22
Coronary heart disease (%)	29.6	29.5	0.97
Peripheral arterial disease (%)	29.3	22.9	<0.001
Peripheral sensory neuropathy (%)	30.8	58.6	<0.001
eGFR (CKD-EPI) category (%)			0.001
≥90 mL/min/1.73m^2^	32.2	38.3	
60–89 mL/min/1.73m^2^	49.8	44.7	
45–59 mL/min/1.73m^2^	11.9	9.1	
30–44 mL/min/1.73m^2^	4.4	5.2	
15–29 mL/min/1.73m^2^	1.2	1.9	
<15 mL/min/1.73m^2^	0.5	0.8	
Urinary albumin:creatinine ratio (mg/mmol)	5.2 (1.5–17.8)	3.3 (0.9–12.9)	<0.001

**Table 2 jcm-09-01428-t002:** Cox and Fine and Gray competing risk regression coefficients for the variables in the five-year cardiovascular disease risk models for three-point major adverse cardiovascular events (MACE) and four-point MACE.

	Model 1: Cox Three-Point MACE	Model 2: Fine and Gray Three-Point MACE	Model 3: Cox Four-Point MACE	Model 4: Fine and Gray Four-Point MACE
Age – 65.7 (years)	0.0213	0.0133	0.0306	0.0273
(Age – 65.7)^2^ (years^2^)	0.0011	0.0009	0.0009	0.0006
Sex (0 = female, 1 = male)				0.2924
Australian Aboriginal (0 = no, 1 = yes)	0.9873	0.9781	0.6854	0.5830
Heart rate – 70 (beats/minute)				0.0173
Diabetes duration – 10.2 (years)			0.0187	0.0162
log_e_(HbA_1c_) – 3.98 (mmol/mol)	0.8371	0.9488	0.7120	0.5898
log_e_(serum total:HDL-cholesterol ratio) − 1.27 (mmol/L)	0.6137			
log_e_(urinary albumin:creatinine ratio) – 1.22 (mg/mmol)	0.5342	0.5307	0.1906	0.1791
eGFR (CKD-EPI) 45–59 mL/min/1.73m^2^	0.5399	0.5936		
eGFR (CKD-EPI) < 45 mL/min/1.73m^2^	0.8599	0.7998	0.6472	0.6559
Peripheral arterial disease (0 = no, 1 = yes)	0.5712	0.6186	0.3071	0.4006
Left ventricular hypertrophy (0 = no, 1 = yes)	1.6355	1.5301	1.0864	1.0617
Heart failure (0 = no, 1 = yes)			0.8803	0.8602
Coronary heart disease and/or cerebrovascular disease (0 = no, 1 = yes)	1.0245	0.9975	0.7203	0.7182

**Table 3 jcm-09-01428-t003:** Performance of five-year cardiovascular disease risk models for three-point major adverse cardiovascular events (MACE) and four-point MACE derived using a) Cox and b) Fine and Gray competing risk regression analyses.

Regression Method	Outcome	Observed N (% (95% CI))	Predicted N (%)	AUC (95% CI)	H-L Test, *p*-Value	Brier Score (Range)	Sensitivity (%) *	Specificity (%) *	PPV (%) *	NPV (%) *
Cox	Three-point MACE	143 (9.2 (7.9–10.8))	161.9 (10.4)	0.77 (0.73–0.82)	<0.001	0.08 (0.00–0.999)	64.3	79.3	24.0	95.6
Fine and Gray	Three-point MACE	143 (9.2 (7.9–10.8))	152.4 (9.8)	0.77 (0.73–0.81)	<0.001	0.08 (0.00–0.999)	62.9	80.0	24.3	95.5
Cox	Four-point MACE	245 (15.8 (14.0–17.7))	231.3 (14.9)	0.81 (0.78–0.84)	0.058	0.10 (0.00–0.96)	81.2	65.2	30.4	94.9
Fine and Gray	Four-point MACE	245 (15.8 (14.0–17.7))	221.2 (14.3)	0.82 (0.79–0.85)	0.17	0.10 (0.00–0.95)	79.2	68.1	31.8	94.6

* for a 10% five-year cardiovascular disease risk cut-off. AUC = area under the receiver operating curve. H-L = Hosmer-Lemeshow. PPV = positive predictive value. NPV = negative predictive value.

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
