# Peer review of "Contemporary Cardiovascular Risk Assessment for Type 2 Diabetes Including Heart Failure as an Outcome: The Fremantle Diabetes Study Phase II"

_jcm, 2020, doi:10.3390/jcm9051428_

Round 1
Reviewer 1 Report
The topic of the paper is of interest, and methods are consistent with the porpoise of the study.
However, in the current form the manuscript is hard to read in some parts, and there is the risk that some messages are lost in the text.
Specific comments:
- I disagree with the idea that ‘locally developed equations should ideally be used’ in risk assessment. Indeed, to make an equation/score useful, it is mandatory that it can be used everywhere in the world. Therefore, please delete all sentences in the text that are referred to this topic. On the other hand, please add in the text that this equation/score has been derived mainly from Australian cohorts, resulting in a possible biases derived from peculiarity of Australian population (e.g. differences in national health system, life styles etc).
- Please, try to make simpler the message of the paper. The main message is that a four MACE points risk score works better than a three MACE points risk score.
- I suggest to change the name of the risk model from CVD1/CVD2 in 3-points/4-points. This makes the message clearer.
- I suggest to change order of results presentation. Paragraph 2.2 and 2.3 can be placed after the presentation of models on CVD1/CVD2.
- I think that the use of 4 different models makes more difficult to use (and compare) results. Please, consider to use only one model for CVD1 and one model for CVD2 (with others as Suppl).
- A clear ‘study limitation’ section needs to be placed at the end of discussion.
- Conclusions are not in line with the main message of the paper. Please, modify
Author Response
Specific comments:
- I disagree with the idea that ‘locally developed equations should ideally be used’ in risk assessment. Indeed, to make an equation/score useful, it is mandatory that it can be used everywhere in the world. Therefore, please delete all sentences in the text that are referred to this topic. On the other hand, please add in the text that this equation/score has been derived mainly from Australian cohorts, resulting in a possible biases derived from peculiarity of Australian population (e.g. differences in national health system, life styles etc).
Response: In the Introduction, we point out in the first paragraph that risk equations may not perform well when used outside the populations in which they were developed, with a supporting reference. We provide further Australia-specific examples in the second paragraph in which risk equations developed in other countries did not perform well in our cohort. However, in view of the Reviewer’s comment, we have amended the First paragraph of the Introduction to read “Reflecting differences in populations, endpoint ascertainment, the predictive variables included and algorithms employed, and changes in management over time, there is considerable variability in their characteristics and performance, while the equations may not perform well when used outside the populations in which they were developed [3]. There is, therefore, an argument that locally developed equations should ideally be used but there may not be relevant contemporary longitudinal data available to allow this.”
We have also, in a new limitations paragraph, commented on the source of the data being from an urban Australian population: “Although the FDS2 cohort was from a typical urban Australian population base and were representative of those with the disease in the study catchment area, their characteristics may differ from those of people with type 2 diabetes in other geographical settings.”
- Please, try to make simpler the message of the paper. The main message is that a four MACE points risk score works better than a three MACE points risk score.
Response: We have, in response to this comment and the fact that that competing risk regression is better than Cox regression, amended the Conclusion of the Abstract to read :” A validated four-point MACE competing risk model reliably predicts key T2D CVD outcomes.”
- I suggest to change the name of the risk model from CVD1/CVD2 in 3-points/4-points. This makes the message clearer.
Response: We have deleted CVD1 and CVD2 and replaced them with three-point MACE and four-point MACE, respectively, throughout the manuscript as requested.
- I suggest to change order of results presentation. Paragraph 2.2 and 2.3 can be placed after the presentation of models on CVD1/CVD2.
Response: We respectfully disagree with this suggestion. Because in order to make the case for an updated CVD risk equation, we needed to show that older ones did not work well in our contemporary FDS2 cohort.
- I think that the use of 4 different models makes more difficult to use (and compare) results. Please, consider to use only one model for CVD1 and one model for CVD2 (with others as Suppl).
Response: We also respectfully disagree with this suggestion since we believe the use of competing risk regression needs to be shown to be superior to Cox regression in predicting CVD risk AND that the 4-point MACE is better than the 3-point MACE.
- A clear ‘study limitation’ section needs to be placed at the end of discussion.
Response: As mentioned above, a new limitations paragraph has been added as suggested.
- Conclusions are not in line with the main message of the paper. Please, modify
Response: As well as the change in the Conclusion in the Abstract, we have also amended the Conclusion in the main text to: “Data and CVD risk equations from FDS1 and FDS2, including previous validations of the Framingham and UKPDS equations [5] and the present ADVANCE risk calculator validation, show that there is a need, where the data are available, to develop local CVD risk prediction equations that are updated as diabetes epidemiology changes. This includes consideration of clinically significant events such as heart failure which may be increasing in incidence. Where local data are unavailable or unsuitable, equations developed in other ethnically similar populations can be applied as long as their potential limitations are acknowledged and the possibility of recalibration explored [11]. The four-point MACE competing risk equation developed from FDS2 data and validated in the independent BDS cohort should replace the Framingham equation-based CVD risk assessment currently recommended for Australians with type 2 diabetes [4] and the possibility of its application beyond an Australian context should be explored.” .
Reviewer 2 Report
It is a very powerful manuscript investigating pertinent clinical factors that might better predict the cardiovascular disease outcomes and cardiovascular disease risks in type II diabetes. The study includes a large number of patients. The study design is correct and feasible. The follow up was well done with a large number of the patients included for a five year follow up. All these lead to a powerful and accurate analyses.
The new phase II study tried to answer new question in addition to their phase I study and tried to acquire a new calculation equation to better predict the cardiovascular disease risks in type II diabetes. The author hypothesized that adding heart failure as a new factor can increase the accuracy of the prediction. It is a good suggestion. The results support this hypothesis.
Minor point:
If the author can further clarify how the heart failure was determined in the study, and which stages of the heart failure (for example, NYHA classification I, II, III or IV) has the most powerful predictive value, it would be even better.
The conclusion in the abstract (line 27) is a little bit vague compared to the powerful results.
Author Response
Minor point:
If the author can further clarify how the heart failure was determined in the study, and which stages of the heart failure (for example, NYHA classification I, II, III or IV) has the most powerful predictive value, it would be even better.
Response: In the Methods we stated that, using International Classification of Disease (ICD)-10-AM coding, the Hospital Morbidity Data Collection, which includes comprehensive data on all public/private hospitalisations in WA, was searched for heart failure from study entry to end-2016. We determined prevalent heart failure similarly as prior hospitalisation for/with heart failure. We agree that this would not have captured heart failure if at levels not requiring inpatient care which we have addressed in the limitations paragraph through the sentence “We used hospitalisations and deaths as key outcomes which would not have captured silent events or symptomatic events that were managed outside hospital.”
The conclusion in the abstract (line 27) is a little bit vague compared to the powerful results.
Response: This has been amended as suggested to “:” A validated four-point MACE competing risk model reliably predicts key T2D CVD outcomes.”
Reviewer 3 Report
The authors compared the predictive accuracy of new risk score in patients with T2D in Australian cohort. They set 2 composite outcomes and developed risk score using Cox and Fine and Gray model. They used derivation and validation cohort which include some Aboriginal. They found that previously published risk score is inferior to their new risk score, and this is dependent on the history of diabetes, ethnic/racial difference and way of current treatment for T2D. This is an interesting and well written paper.
Author Response
No response required
Round 2
Reviewer 1 Report
The paper has been ameliorated.
Please, undertone a bit the sentence in the conclusion paragraph ('the independent BDS cohort should replace the Framingham equation-based CVD risk assessment 463 currently recommended for Australians with type 2 diabetes').
Author Response
Please, undertone a bit the sentence in the conclusion paragraph ('the independent BDS cohort should replace the Framingham equation-based CVD risk assessment 463 currently recommended for Australians with type 2 diabetes').
We have replaced ‘should replace’ with ‘could be considered as a replacement for’ to 'soften' the recommendation.